# Outcomes and Challenges in Noncommunicable Disease Care Provision in Health Facilities Supported by Primary Health Care System Strengthening Project in Sri Lanka: A Mixed-Methods Study

**DOI:** 10.3390/healthcare11020202

**Published:** 2023-01-09

**Authors:** Divya Nair, Pruthu Thekkur, Manoj Fernando, Ajay M. V. Kumar, Srinath Satyanarayana, Nadeeka Chandraratne, Amila Chandrasiri, Deepika Eranjanie Attygalle, Hideki Higashi, Jayasundara Bandara, Selma Dar Berger, Anthony D. Harries

**Affiliations:** 1Centre for Operational Research, International Union Against Tuberculosis and Lung Disease (The Union), 75001 Paris, France; 2Department of Health Promotion, Rajarata University of Sri Lanka, Mihintale, Anuradhapura 50300, Sri Lanka; 3The Union-South East Asia (USEA) Office, New Delhi 110016, India; 4Yenepoya Medical College, Yenepoya (Deemed to be University), Mangalore 575018, India; 5The Foundation for Health Promotion, 21/1 Kahawita Road, Dehiwala 10350, Sri Lanka; 6Department of Community Medicine, Faculty of Medicine, University of Colombo, Colombo 00300, Sri Lanka; 7The World Bank, Colombo 00300, Sri Lanka; 8Project Management Unit, Primary Health Care System Strengthening Project (PSSP), Colombo 00300, Sri Lanka; 9Department of Clinical Research, Faculty of Infectious and Tropical Diseases, London School of Hygiene and Tropical Medicine, London WC1E 7HT, UK

**Keywords:** noncommunicable diseases, screening, continuity of patient care, perspectives, primary healthcare, health system, operational research

## Abstract

The Primary Healthcare System Strengthening Project in Sri Lanka focuses on improving noncommunicable disease (NCD) care provision at primary medical care institutions (PMCIs). We conducted an explanatory mixed-methods study to assess completeness of screening for NCD risk, linkage to care, and outcomes of diabetes/hypertension care at nine selected PMCIs, as well as to understand reasons for gaps. Against a screening coverage target of 50% among individuals aged ≥ 35 years, PMCIs achieved 23.3% (95% CI: 23.0–23.6%) because of a lack of perceived need for screening among the public and COVID-19-related service disruptions. Results of investigations and details of further referral were not documented in almost half of those screened. Post screening, 45% of those eligible for follow-up NCD care were registered at medical clinics. Lack of robust recording/tracking mechanisms and preference for private providers contributed to post-screening attrition. Follow-up biochemical investigations for monitoring complications were not conducted in more than 50% of diabetes/hypertension patients due to nonprescription of investigations by healthcare providers and poor uptake among patients because of nonavailability of investigations at PMCI, requiring them to avail services from the private sector, incurring out-of-pocket expenditure. Primary care strengthening needs to address these challenges to ensure successful integration of NCD care within PMCIs.

## 1. Introduction

Strengthening of the primary healthcare (PHC) system to tackle the increasing morbidity and mortality associated with noncommunicable diseases (NCDs) is prominently featured in recommendations of various globally recognized forums [1,2,3,4,5]. This involves reorganizing the PHC system to deliver community-based NCD risk factor preventive services, diagnostic services for early detection of NCDs, and a continuum of quality care for those with NCDs [6,7]. The continuum of care at PHC includes linking NCD patients to appropriate treatment, follow-up on a regular basis for their lifetime, and early detection and management of life-threatening complications through well-established referral systems, allowing referral to secondary or tertiary care facilities and back referral to primary care when complications have been effectively managed [7]. 

A recent review reported that such integration of NCD care and reorientation has been beneficial, not only for NCD care provision but also for improving overall quality of care delivered through the PHC system [8]. However, studies from low- and middle-income countries (LMIC) assessing the integration of NCD care in PHC have shown critical gaps in available infrastructure, trained manpower, and services compared to the standards prescribed in the policies [9,10,11,12,13,14,15,16]. The reasons for such gaps are generally localized and required detailed assessment for working out the best strategies to enhance NCD care provision in PHC [17]. Therefore, the World Health Organization (WHO) recommends annual monitoring of service availability and delivery for supporting policymakers in planning appropriate sustainable responses to ensure quality care provision through PHC [18]. 

Sri Lanka, an LMIC in the South Asia region, is grappling with a high burden of NCDs. These account for 81% of the total deaths and 77% of disability-adjusted life years (DALYs) [19,20]. Although steps were taken to integrate early detection and management of NCDs through the primary medical care institutions (PMCIs), there was underutilization of NCD care services (diagnosis and treatment) [21]. The health information systems in PMCIs lacked the capacity to identify, manage, and track patients over time. Furthermore, preference for seeking NCD care at secondary and tertiary hospitals and in the private sector resulted in overutilization of higher-level healthcare and increased out-of-pocket expenditure (OOPE) [22]. All these together led to a lack of continuous, coordinated, and patient-centered NCD care [21]. 

To overcome existing deficiencies in care provision at PMCIs, the Ministry of Health (MOH) implemented the Primary Healthcare Systems Strengthening Project (PSSP) in 2018 [23]. PMCIs include all district level hospitals and primary medical care units. The MOH implements the project with the financial support from the World Bank, with payments made annually on the basis of achievement of results against agreed disbursement-linked indicators (DLIs). There are nine DLIs with targets set for each year [24]. The PSSP supports defining and providing quality medical care at PMCIs, innovating and integrating prevention and treatment for NCDs through healthy lifestyle centers (HLCs) and streamlining referrals. It also supports establishment of the health management information system (HMIS), trained manpower, efficient supply chain management systems, and laboratory service capacity [23]. 

Two years into the implementation of PSSP, we conducted the first systematic assessment of the status of implementation of the project in the country to generate evidence for guiding its future scale-up. We used quantitative and qualitative techniques of inquiry to assess facility-level infrastructure for care provision, coverage of empanelment, process of NCD care provision, and patient experience at PMCIs supported by PSSP. In this paper, we report the findings of assessment of the coverage and completeness of screening for NCD risk factors, linkage with care and outcomes of care among those enrolled for diabetes and/or hypertension care at nine PMCIs in Sri Lanka. We complemented the quantitative assessment with an exploration of providers’ and patients’ perspectives regarding the challenges in NCD care. As suggested by the WHO, findings from such an assessment could enable program managers to find context-specific solutions to fix gaps and further improve the NCD care in primary care settings [17].

## 2. Materials and Methods

### 2.1. Study Design

This was an explanatory mixed-methods study [25]. The quantitative component was a longitudinal descriptive study involving secondary data extraction from facility and patient records. Each quantitative assessment was followed by a qualitative component, which was a descriptive study with in-depth interviews among healthcare workers and patients.

### 2.2. Study Setting

#### 2.2.1. General Setting

Sri Lanka is an island country with a population of 21.8 million as of 2019 [26]. Administratively, the country is divided into nine provinces, each governed by an autonomous provincial council. The provinces are further divided into districts, administered by a district secretariat [27]. The most peripheral local administrative units are Grama Niladhari (GN) divisions [28].

#### 2.2.2. Specific Setting

In Sri Lanka, curative services are provided at three levels in the public health system: primary (through PMCIs), secondary (base hospitals), and tertiary (district hospitals and higher facilities) [29]. The PSSP focuses on reorganizing the PHC system by strengthening PMCIs to become capable of providing patient-centered curative services, especially NCD care. The project supported the establishment of physical infrastructure including laboratory testing facilities and a laboratory network system, provision of medical equipment, introduction of population empanelment including screening for NCD risk factors, and strengthening NCD care provision in PMCIs [23]. The key features of the reorganization include the following:Empanelment of the population: Through this process, the population of a given GN division is assigned to a PMCI. Each individual in the identified catchment population is assigned a unique personal health number (PHN), and a personal health record (PHR) is created. The PHR includes information on the clinical profile at the time of registration and needs to be updated at every encounter with a healthcare provider. Currently, the PHR is a paper record with the unique identifier (PHN) made available to patients at the time of registration. The PHR is available with the patient and brought for each facility visit.Screening: During empanelment, the PMCIs actively screen individuals aged ≥35 years for intermediate risk factors for NCDs such as raised blood pressure (BP), raised blood glucose levels, abnormal blood lipids, overweight, and obesity, through either outreach activities or opportunistic screening at facilities. Laboratory investigations as part of the screening are expected to be available at the PMCI or through an external laboratory (sample collection and transfer linkage with PMCI). The demographic details of all the individuals screened, the results of investigations conducted, and the unit referred for further follow-up are documented in the “participants register” maintained at the PMCI. The 1 year and 2 year targets were to screen at least 25% and 50% of individuals aged 35 years and more, respectively [23].Follow-up and referral: Those with a low level of cardiovascular disease (CVD) risk and no intermediate risk factors are followed up at HLCs for risk reduction through lifestyle modification. Individuals followed up in the HLC are provided with a card similar to the PHR, which should note progress on the detected risk factors. The details of follow-up are recorded in the paper-based “HLC follow-up register”, in addition to being updated in the HLC module of electronic PHR. At least 25% of those enrolled at HLC are expected to have made a minimum of one follow-up visit within 1 year of the initial visit. The HLCs receive services from health promotion officers, counsellors, dieticians, instructors in physical education, and nutritionists.

Individuals with either high BP, high blood glucose, high serum creatinine, or 10 year CVD risk ≥ 30% are referred to the medical clinic at the PMCI for further management. These patients diagnosed at PMCI or reaching the PMCI having been diagnosed elsewhere are registered for care at the medical clinics and listed in the “medical clinic register”. All details on further clinical care delivered to the patient are documented in the PHR. The services mentioned above are provided in accordance with standard guidelines issued by the MOH [30]. 

### 2.3. Study Population

#### 2.3.1. Quantitative Component

##### Assessment of Care Cascade

All the individuals screened for NCD risk factors from June 2019 to June 2021 and listed in the “participant register” of the nine selected PMCIs were included for assessing the care cascade. 

##### Assessment of Quality of Follow-Up Care

All the individuals registered at any time for diabetes and/or hypertension care at the medical clinic in the selected PMCIs were eligible for the study. As this assessment was conducted telephonically, only those individuals with a valid mobile phone number documented in the clinic register were included. Those individuals who confirmed that they had diabetes and/or hypertension and consented to participate in the study when contacted over the phone were included, up to 150 patients per PMCI.

#### 2.3.2. Qualitative Component

Medical officers (n = 4), nursing officers (n = 2), program managers (2), and patients availing NCD care (n = 8) at four PMCIs were interviewed to understand the process and challenges of screening and providing NCD care. The four PMCIs from four different provinces were selected on the basis of ease of access to the research team amidst COVID-19-related mobility restrictions (e.g., lockdown) imposed in the country. The medical officers in charge of each selected PMCI were interviewed. Nursing officers in charge of NCD care provision, who were available at the PMCIs on the day of visit by the study team, were also interviewed. One program manager from each of the four provinces was requested to provide interviews, but only two were able to provide time for participation. The patients who were available on the day of the visit to the PMCI and for whom the follow-up investigations were not performed as per guidelines were approached for interview.

### 2.4. Data Collection, Study Variables, Data Source, and Study Tools

#### 2.4.1. Quantitative Component

##### Assessment of Care Cascade

We extracted information on demographic characteristics, behavioral risk assessment, anthropometric measurements, biochemical investigations, 10 year CVD risk prediction, and referral for further care (HLC or medical clinic) from the HMIS. Although the HMIS is functional in all the PMCIs, eight out of nine PMCIs had not recorded screening details for all the individuals screened on the HMIS during the study reference period. Thus, we extracted data from the paper-based “participants register” maintained at the PMCIs. 

Parameters such as body mass index, BP, and blood glucose were recorded as categorical variables in an older version of registers at some PMCIs while the newer versions required these to be recorded as quantitative continuous variables. The definitions followed in Box 1 were employed to make raw data extracted from the different versions of the registers comparable of each other.

Box 1Cutoffs and definitions for parameters used in deducing indicators for NCD care provision in the nine selected PMCIs supported by the PSSP in Sri Lanka, 2021.

*Cutoffs and definitions for parameters used during screening for NCD risk factors*



**Body mass index (BMI):**
New version: <18.5 kg/m^2^, underweight; 18.5 to 24.9 kg/m^2^, normal; 25 to 29.9 kg/m^2^, overweight; ≥30 kg/m^2^, obeseOld version: The categories as assigned in the register were retained, i.e., entries recorded as “underweight” were considered underweight and so on.

**High blood pressure (BP):**
New version: systolic blood pressure ≥ 140 mmHg or diastolic blood pressure ≥ 90 mmHgOld version: entries recorded as “above or equal to 140/90 mmHg” or “above or equal to 160/100 mmHg”

**High blood glucose:**
New version: fasting blood glucose ≥ 126 mg/dL or random blood glucose ≥ 200 mg/dLOld version: entries recorded as “fasting blood glucose ≥ 126 mg/dL”

**High total cholesterol: ≥200 mg/dL**

**High serum creatinine: >1.2 mg/dL**



*Indicators assessing process of care in monitoring of complications in individuals with diabetes and/or hypertension*


BP measurement taken in the last visit to PMCIBlood glucose test conducted in last 3 monthsLipid profile taken (either total cholesterol alone or complete lipid profile) in the last yearRenal function test conducted (either serum creatinine or serum urea) in the last yearElectrocardiography (ECG) performed in the last yearFundus examination conducted in the last yearFoot examination conducted in the last year


*Indicators assessing the status of disease control*


BP control: individuals with systolic blood pressure < 140 mmHg and diastolic blood pressure < 90 mmHg on BP measurement during the last visit to PMCIBlood glucose control: individuals with fasting blood glucose < 126 mg/dL or random blood glucose < 200 mg/dL in the last 3 monthsLipid profile within normal limits: individuals with total cholesterol < 200 mg/dL, LDL < 100 mg/dL, HDL ≥ 40 mg/dL, and triglycerides < 150 in the last yearRenal function within normal limits: individuals with serum creatinine < 1.3 mg/dL in the last year


Linkage to medical clinic: The HMIS module of the medical clinic was available in all the PMCIs, but was not being used in all the PMCIs. Although the clinic registers were maintained, the details such as PHN, date of registration, type of NCD, and pathway for registration were incomplete and did not allow us to carry out cohort-like linkage of individuals from screening to initiation of treatment at medical clinics. Hence, we counted the total number of individuals registered in the medical clinic since June 2019. This aggregate number of registered individuals at the medical clinic might have included persons registered for care of other NCDs (e.g., asthma, osteoarthritis, and geriatric care), as well as those not detected through screening by the PMCI. In two of the nine PMCIs, on each day of the clinic, the staff listed the details of individuals attending the clinic in the clinic register without indicating whether the attendee was a new or an old registration. Thus, it was not possible to extract the aggregate number of individuals registered for care after June 2019.

Adherence to follow up visits at HLC: The number of individuals who made at least one follow-up visit at HLC since June 2019 was extracted from the HMIS and/or paper-based “HLC follow-up” register. Although the HLC module of HMIS was available in all PMCIs, the follow-up details were not routinely updated on the HMIS module but were documented in the paper-based “HLC follow-up” register. In five out of nine PMCIs, there was no paper-based “HLC follow-up” register. According to the staff, the details of the follow-up visit were documented in either the PHR of the patient or the participant registers when screening investigations were repeated. Hence, the participants’ register was digitized, and the duplicates were removed to get the total number of individuals with at least one HLC follow-up visit after being screened initially. In these situations, the total number of individuals with duplicates in the “participant register” were assumed to be those with at least one follow-up visit after the initial screening visit.

##### Assessment of Quality of Follow-Up Care

The participants were either contacted at the clinic or requested to share images of their clinic book or the PHR for data extraction. Demographic details, clinical and biochemical parameters, types of health facilities visited, and number of visits made to PMCI in the last year from the date of data extraction were extracted from the clinic book or PHR as per a checklist (see Appendix A).

Compliance with follow-up clinical assessments: The eligibility for certain investigations/examinations is based on the duration for which the individuals have been registered with the PMCI. For this study, the duration between the date of registration at the clinic and the date of data extraction was used as the reference period for assessing eligibility for investigations. Indicators for assessing the compliance with guidelines on monitoring for complications and status of disease control in individuals with diabetes and/or hypertension are shown in Box 1.

#### 2.4.2. Qualitative Component

All the in-depth interviews were conducted by the research consultants who were medical doctors familiar with the health system in Sri Lanka, fluent in local languages (Sinhala and Tamil), and trained and experienced in qualitative research. Separate sets of interview schedules (see Appendix A) with probes were used for interviews among healthcare providers and patients. The interviewers were guided by the quantitative findings to specifically understand the identified gaps. For example, the proportion of individuals screened in their PMCI was presented to the healthcare provider (respondent) to make them narrate the challenges in light of this information. The interviews were audio-recorded and were used to prepare the transcripts. On average, the duration of the interviews was 13.06 min (range 6.01–43.00 min).

### 2.5. Data analysis

#### 2.5.1. Quantitative Component

##### Assessment of Care Cascade

The data extracted from the “participant register” were entered directly into an EpiCollect5 application (Wellcome Sanger Institute, Cambridge, UK), a mobile phone-based data capture tool. Stata version 12 (StataCorp LP, College Station, TX, USA) was used for analysis. Duplicates were removed using PHN, national ID and the name of the individual to get the total number of individuals who underwent screening in each PMCI. Trends in the aggregate number of screenings performed per month in all the PMCIs during June 2019 to June 2021 were described. Proportions with 95% confidence intervals were used to estimate the prevalence of tobacco smoking, alcohol use, high BP, high blood glucose, high cholesterol, and CVD risk of ≥30% among those with documentation of screening details (excluding those missing from the denominator). The process of linkage to care was summarized using the following percentages:Percentage eligible for follow-up at the medical clinic among those screened,Percentage indicated as referred to the medical clinic in the participant register,Percentage registered in the medical clinic calculated with the aggregate number registered in the medical clinic as the numerator and the total eligible for referral as the denominator,Percentage with at least one follow-up visit at HLC calculated with the aggregate number of individuals making at least one visit as the numerator and the individuals referred to HLC as the denominator.

##### Assessment of Quality of Follow-Up Care

The median (IQR) number of the visits in the last year was calculated. The number and percentage of eligible individuals undergoing BP measurement, blood glucose test, lipid profile, renal function test, fundus examination, foot examination, and ECG as per criteria and with values under control was determined.

#### 2.5.2. Qualitative Component

The transcripts were prepared on the same day of the interview with the use of notes and audio-recording. Thematic analysis was performed by the PI (P.T.K.) using Atlas-Ti software on the challenges of delivering quality NCD care (including screening and establishing linkages and follow-up). The second investigator reviewed the analysis and decisions on coding and categorization were done in consensus. Similar codes were combined into categories. To ensure that the results reflected the data, the codes/themes were related back to the original data. The findings were reported in accordance with “Consolidated Criteria for Reporting Qualitative Research” guidelines [31].

We deduced 63 codes related to challenges in delivering NCD care from the transcripts of the interviews among healthcare providers and patients. The codes were combined to form 15 categories, which were grouped and summarized as challenges in screening for NCD, linkages to the clinics, and follow-up and NCD care provision at the clinics.

## 3. Results

The results are presented under three broad headings representing the steps in NCD care provision (screening, linkage to care, and follow-up). For each step, we present the gaps identified quantitatively, followed by an exploration of possible reasons for these gaps which were identified using qualitative methods.

### 3.1. Screening for NCD Risk Factors and Outcomes

#### 3.1.1. Coverage of Screening (Quantitative Component)

As per the census, there were 86,615 individuals aged 35 years and above in the GN divisions allotted to the assessed nine PMCIs. Of these, according to the paper-based registers, 20,215 (23%, 95% CI: 23.0–23.6%) individuals were screened for NCD risk factors as of June 2021 (2 years from PSSP implementation) (Table 1). Only one out of the nine PMCIs achieved the target of screening > 50% of individuals above 35 years of age.

As shown in Figure 1, a total of 22,697 individuals (including those below 35 years) were screened for NCD risk factors between June 2019 and June 2021. The number of individuals screened was highest (3025) in December 2019 and reduced gradually in 2020. In 2020, the number screened was lowest during April and May 2020, when the COVID-19-related mobility restrictions were in place. In 2021, the number of individuals screened remained low and further declined in April/May 2021 when the countrywide lockdown was instituted in response to an increase in COVID-19 cases.

#### 3.1.2. Completeness of Screening (Quantitative Component)

Of the total number of individuals screened, 4.1% were <35 years of age; in another 4.1%, the age was missing in the participant register. Of those screened, 29.7% were males, indicating a lower proportion of males among those screened compared to the 48% of males among individuals aged ≥ 18 years in Sri Lanka. The details on behavioral risk factors such as tobacco smoking and alcohol use were “not recorded” in 7.5% and 7.0%, respectively. The BMI was “not recorded” in 0.6% of the total number screened. Waist circumference was missing in 20.6%. The percentage of “not recorded” for blood pressure measurement was less than 2% in eight out of nine PMCIs. Blood glucose tests were “not recorded” in 1.8% of the screened. Total cholesterol was “not recorded” in 49.9% of those screened largely due to the inability to conduct the investigation due to lack of adequate supplies of cholesterol strips and poor processes for sample collection and transportation. The CVD risk was not ascertained and recorded in 17.3% of those screened. The level of completeness of screening for different risk factors varied across the PMCIs (Table 2).

#### 3.1.3. Prevalence of Risk Factors (Quantitative Component)

The prevalence of NCD risk factors among those screened with results documented in the “participant register” (excluding “not recorded”) stratified by age, gender and PMCI is given in Appendix A. The prevalence of behavioral risk factors such as tobacco smoking and alcohol consumption among those screened was 7.8% (95% CI: 7.5–8.2%) and 13.3% (95% CI: 12.8–13.7%), respectively. The prevalence of obesity (BMI ≥ 30) among those screened was 11.7% (95% CI: 11.3–12.1%). The prevalence of high BP (BP ≥ 140/90 mmHg) among those screened was 27.4% (95% CI: 26.8–28.0%). The prevalence of abnormal biochemical parameters such as high blood glucose (FBG ≥ 126 mg/dL or RBG ≥ 200 mg/dL), high cholesterol (total cholesterol ≥ 200 mg/dL), and high serum creatinine (>1.3 mg/dL) among those screened was 13.6% (95% CI: 13.1–14.0%), 43.1% (95% CI: 42.2–44.0%), and 8.1% (95% CI: 7.2–9.0%), respectively. The prevalence of CVD risk ≥30% was 1.7% (95% CI: 1.5–1.9%) among individuals in whom the CVD risk status was documented.

#### 3.1.4. Challenges in Screening for NCDs (Qualitative Component)

The challenges highlighted by the healthcare providers and patients when asked about reasons for low coverage of screening and data incompleteness in screening process are described below and depicted in Figure 2.

##### Lack of Perceived Need for Screening (Qualitative Component)

The healthcare workers (HCWs) felt that the general public did not perceive the importance of the screening even when it was explained to them, especially those who felt apparently healthy.


*“I feel that they may not have perceived the true value of the screening process. And I’m not quite sure when they’re invited through the phone, whether the importance of the screening was clearly conveyed to them… And then for people who are not with any illnesses. I mean they don’t get something big in return.”—Medical officer*


##### Nonavailability of Certain Investigations (Qualitative Component)

The HCWs complained that some recommended investigations such as serum creatinine and total cholesterol were not available during the screening activities. Sample collection and transportation mechanisms for receiving tests in the apex laboratories were not adequately established.


*“Initially, we could perform only a few tests. We had only few instruments to conduct the tests. We only had facilities to test fasting blood sugar. We measured height, weight, and blood pressure, and we could conduct breast examination too. But people visited to have their cholesterol levels checked and they needed to undergo kidney tests, which we did not have”—Medical officer*


The HCWs also mentioned that, even when the laboratory facility was available within PMCIs, they were not able to receive all the investigations due to nonavailability of reagents.


*“The other thing is the issue in the laboratories; we have a laboratory complex and adequate staff as well. But some investigations are not performed here as reagents are not there.”—Medical officer*


The nonavailability of investigations was reiterated by patients as well. They also mentioned that they were referred to private laboratories for some investigations, leading to OOP expenditure.


*“After knowing this (high total cholesterol based on a rapid test with cholesterol strips), the doctor gave me a chit so I could get full cholesterol (probably lipid profile) checked from an outside clinic. So, I got it performed by XX hospital (private hospital) and came back.”—Patient*


##### Requirement of Repeat Visits (Qualitative Component)

Patients complained that sometimes results were not readily available on the day of screening, and they had to visit the PMCI again to collect the results. The HCWs also felt some of the patients did not turn up on a second occasion to collect the results and eventually would not be put on treatment in spite of having some abnormalities.


*“When checking glucose, we take ‘blood’, we check and then we inform the patients immediately about the results. But we cannot do this for cholesterol (as it needs to be performed in an apex laboratory through sample collection and transportation). Then, there is the probability of missing a few patients. That means, even though the patient participated in the screening activity, they might not like to spend another day to come and collect results due to personal reasons.”—Medical officer*


##### COVID-19-Related Challenges (Qualitative Component)

The HCWs felt that screening activities reduced drastically with the onset of the COVID-19 pandemic. This was due to a shift in focus of HCWs toward the COVID-19 response, as well as due to the inability to conduct outreach activities for screening. The mobility restrictions reduced the utilization of OPD services, and not many individuals were available for opportunistic screening. Some of the PMCIs were converted into COVID-19 hospitals with suspension of all other services.


*“But I think due to this COVID-19 situation, services have gone down. Last year, in the fourth quarter, I went for supervision, and only nine patients were screened. And they said due to the COVID-19 situation, we also can’t do much. But this year I told them to screen, and, in the first quarter, they screened more than 50 patients”—Program manager*


### 3.2. Linkage to Care

#### 3.2.1. Documentation of Referral, Eligibility, and Registration at Referred Clinics (Quantitative Component)

Of the 22,697 individuals screened, 14,486 (63.8%) and 8211 (36.2%) were eligible for referral to HLC and a medical clinic for further follow-up, respectively. Of the total number screened, the place of referral for further follow-up (either HLC or medical clinic) was “not recorded” in 10,288 (45.3%) individuals. Of the 14,486 eligible for referral to HLC, in only 7152 (49.4%) was there documentation of referral to HLC. Of those eligible for referral to follow-up at HLC, 682 (4.7%) were referred to the medical clinic for further follow-up. Of the 8211 individuals eligible for referral to the medical clinic, 2607 (31.7%) were wrongly referred to HLC for further follow-up and only 1968 (24.0%) had documentation of referral to the medical clinic in the “participant register” (Figure 3).

In total, 1199 individuals made at least one follow-up visit to HLC between June 2019 and May 2021 after being screened once for NCD risk factors, and this amounted to 8.3% of the total individuals referred to HLC for further follow-up (Figure 4). Between June 2019 and May 2021, 3352 individuals were registered at the medical clinic of the seven PMCIs for further follow-up. In two PMCIs, the data on the number of individuals registered during the study reference period were not available. Of the 7466 individuals eligible for follow-up at the medical clinic from the seven PMCIs (8211 from all nine PMCIs), the percentage of those registered for medical care was 3352 (44.9%). The calculated percentage might be an overestimate of those registered as the aggregate number of registered individuals extracted from the clinic register may include the individuals diagnosed with other NCDs (e.g., asthma, osteoarthritis, and geriatric care), as well as those not detected by the screening through the PMCI.

#### 3.2.2. Challenges around Linkages to Clinic and Follow-Up (Qualitative Component)

The challenges in linking the screened individual to either HLC or the medical clinic, according to the screening results, and the challenges in ensuring regular follow-up visits of individuals to these clinics are described below and depicted in Figure 2.

##### No Systems to Track Linkages and Follow-Up (Qualitative Component)

The HCWs felt that the systems for tracking with either a paper-based register or electronic HMIS were not well established for tracking the individuals once they were screened for NCD risk factors. The PHRs were available only with the patients, and electronic PHRs were not available. Thus, there was no opportunity to derive a list of individuals not linked at medical clinic after screening. Similarly, there were no registers to document the registration and follow-up details of patients seeking care in the medical clinic.


*“Actually, we are not maintaining registers for documenting follow-up details of patients followed up in the medical clinic. We will do it in future. Once medical clinic details are entered in the HMIS, we will be able to track the patients. We have the register for the second HLC follow-up visit; we need to enter it into HMIS. We will do it. All these are because of COVID-19”—Medical officer*


The patients also felt that there was no provider initiated active tracking for follow-up at HLC after 1 year of initial screening.


*“They said that there would be a review examination after 1 year, when I was screened in 2019. But they didn’t call me thereafter.”—Patient*


##### Seeking Care in Private Clinics after Screening (Qualitative Component)

The HCWs mentioned that some of the individuals preferred to seek care in the private clinics after being diagnosed with diabetes or hypertension during screening, thus leading to patients being lost-to-follow up from PMCI care after screening for NCD risk factors.


*“Yes, then, they go to the private clinic. These are practical issues; they (patients) get frightened about their blood glucose and go for private treatment. Then, they go XXX (bigger private hospital) or somewhere continuously. That is their choice.”—Medical officer*


The HCWs mentioned that individuals who were affluent and educated preferred to seek care from private clinics due to concerns about the quality of the drugs which were disbursed as loose pills instead of blister packs through the PMCIs.


*“I feel that most of the people who go to the private sector are people who are working and educated, so they look for the quality of the drugs that are issued. Now, when it comes to the drugs that we issue, we know we give them loose pills. We don’t even give them in a blister pack. And some people always go for the brand names or the quality of the drug, and they’re not happy with the type of the drugs that we give at our hospitals.”—Medical officer*


The HCWs mentioned that patients moved to private clinics due to the indifferent attitude of the HCWs in the PMCIs. Some of the HCWs behaved rudely and never greeted the patients.


*“People are not very much welcomed in the government system, I mean, when a patient arrives, in the first few minutes. We are not like the private sector, only in a few hospitals (government hospitals) does the attendant (HCWs) at least look at the patient and pleasantly smiles at the patient, saying ‘Good Morning!’ There is no such greeting system… Most of our people (staff members) are behaving like they have a lot of work and a burden on their minds.”—Medical officer*


Some HCWs and patients mentioned that it was considered fashionable and prestigious to seek care in the private sector by spending a significant amount of money.


*“Some people go to the private sector just because they have money, and, because they are high and mighty, they go to private places without appreciating the dispensary.”—Patient*


##### Early Dropouts with Clinical Improvement (Qualitative Component)

The HCWs complained that some patients, who had successfully registered in the medical clinic for care, stopped visiting the clinic once there was some clinical improvement.


*“You know there are dropouts. For example, some people start with a BMI of 30 or 40. Blood pressure too needs to be corrected. Then, sometimes blood glucose needs correction. So, they come to clinics for 2 or 3 months and then when things become a bit smoother, they forget about the pills and don’t come to the clinic”—Medical officer*


##### Lost to Follow-Up due to Migration (Qualitative Component)

One of the reasons for not being linked to medical clinics after screening and being lost to follow-up after registering at the medical clinic was outmigration from the catchment area.


*“No. I couldn’t. I was away from here (staying in city) for several months because of my mother’s illness. They had given me a scheduled date (for follow-up visit) during that time, and I couldn’t go on that date.”—Patient*


### 3.3. Assessment of Quality of Follow-Up Care at Medical Clinic

#### 3.3.1. Regularity of Follow-Up Visits and Monitoring for Complications (Quantitative Component)

Of the 1154 individuals registered in the medical clinic and selected for assessment of quality of follow-up care, we were able to contact 818 (70.8%) of the individuals, and 791 (96.7%) of them consented to take part in the study. Of the 791 who consented, 675 (85.0%) had diabetes and/or hypertension. Among 675 individuals included in the study, data were available and extracted from the PHR in 26 (3.9%) patients. In the remainder, the data were extracted from the clinic book or the register.

Of the 675 individuals, 217 (32.1%) had only diabetes, 326 (48.3%) had only hypertension, and 132 (19.6%) had both diabetes and hypertension. Of the total, the mean (standard deviation) age of the participants was 58.6 (11.2) years, and 472 (71.4%) were females. Among all individuals, 454 (67.3%) were registered 12 months prior to the date of data extraction, whereas 44 (6.5%) had registered within 3 months of data extraction. The median (IQR) number of visits made by the participants to the PMCI in the last 1 year prior to assessment was six (4–9). The demographic and clinical details of the participants stratified by the type of NCD are shown in Appendix A.

For all the further analysis on the quality of care, individuals with “only diabetes” and those with “both diabetes and hypertension” were grouped as “individuals with diabetes”. The indicators on the quality of care were deduced among those with diabetes and those with “only hypertension”.

Of the 349 individuals with diabetes (217 with only diabetes and 132 with diabetes and hypertension), 199 (57.0%) underwent blood pressure measurements during their last visit to the medical clinic, of which 126 (63.3%) had their blood pressure under control. Of the 328 individuals eligible for follow-up blood glucose test, the tests were performed in 117 (35.7%) at least once in the last 3 months, and 46 (39.3%) had their blood glucose under control. Of the 229 individuals eligible for lipid profile and renal function tests, these tests were conducted in 41 (17.9%) and 16 (7.0%), respectively. The proportion who underwent ECG (4.4%), foot examination (1.8%), and fundus examination (0.9%) in the last year was less than 5% (Table 3).

Of the 326 individuals with only hypertension, 228 (69.9%) underwent blood pressure measurement during their last visit to the medical clinic, of which 107 (46.9%) had their blood pressure under control. Of the 229 individuals with only hypertension registered for care 12 months prior to the date of data extraction (eligible), 62 (27.6%) had their blood glucose tested at least once in the last year. Similarly, 28 (12.4%) had their lipid profile tested and seven (3.1%) had their renal function tested at least once in the last year. The proportion of those eligible who underwent ECG (5.8%) and fundus examination (0.4%) in the last year was low (Table 3).

Of the 179 individuals who underwent blood glucose testing as per the guidelines, 78 (43.5%) were tested at the PMCI itself, and 26 (14.5%) availed investigations from other public health facilities. About 51 (34.2%) had their blood glucose tested in a private laboratory or hospital, paying for the service. Similarly, more than one-fourth of the individuals undergoing lipid profile (37.8%) and renal function test (30.4%) availed these services from a private facility. All the individuals who underwent foot and fundus examination received the service from the PMCI (Table 4).

#### 3.3.2. Challenges around NCD Care Provision at Clinics (Qualitative Component)

The challenges highlighted by the healthcare providers and patients when asked about NCD care provision through medical clinics are described below and depicted in Figure 2.

##### Inability to Deliver Service due to COVID-19 (Qualitative Component)

The HCWs unanimously felt that social distancing requirements imposed during the pandemic precluded any assessment requiring close proximity with the patient.


*“I can’t ask my staff to draw blood or do investigations due to COVID-19. Even in those who are having serious illnesses in hospitals, even the blood pressure is not measured because of the fear of COVID-19. In that context, how can I make my staff go very close to people and do it? I mean, that is a risk.”—Medical officer*


Furthermore, the patients said that, due to COVID-19 and the mobility restrictions with lack of transportation facilities, they were not able to visit the medical clinic to collect their monthly drugs.


*“It was during the curfew (COVID-19 related lockdown), and three-wheelers were charging double the fare. So, I just visited a private medical center near my house.”—Patient*


##### Lack of Space and Equipment at PMCI (Qualitative Component)

The HCWs complained that the PMCIs lacked the required infrastructure for conducting health education sessions or for training the patients in common exercises to promote physical activity for NCD patients.


*“There are people who are willing to do exercise while they are here. There should be a place for that and there should be a ‘washroom’. Further, if any small lecture is given, there should be a small space for that. Actually, the place should be built by keeping all these issues in mind. In the currently available clinic space, we cannot do those things. At a time, we have to arrange chairs to do that and then we need to rearrange the chairs. That is a waste of time. The place should be well designed.”—Medical officer*


##### Shortage of Drugs and OOP Payment for Drugs (Qualitative Component)

The patients highlighted the shortages of NCD drugs in the PMCIs, such that they had to purchase the drugs from the private pharmacies with OOP expenditure.


*“Sometimes in the past few months some medicines were not available. It was for around 4–5 months. I bought medicine for diabetes and high blood pressure from the pharmacy. So, there were issues like this.”—Patient*


##### Poor Monitoring for Complications (Qualitative Component)

Almost all patients, when asked whether they underwent follow-up investigations for monitoring the complications associated with NCDs, said they were not aware that they had to undergo such investigations and/or the doctors had not prescribed them.


*“No, they did not do any test after detecting high blood pressure, they did not tell me anything (to get the follow-up investigations). Dr XXXX (family doctor) asked me to do them (blood investigations—blood sugar, serum creatinine, and cholesterol) 4 months back.”—Patient*



*“No, I did not check them here (registered PMCI). I went to XXX hospital (secondary hospital). I went myself there around 2 years ago and got it (fundus examination) checked.”—Patient with diabetes and on treatment for the last 5 years*


While the patients said the investigations were not prescribed, the HCWs mentioned that patients did not receive investigations in spite of the HCWs prescribing and requesting the patients to receive them.


*“We are asking patients to come for investigations, but if they don’t come, we can’t do anything.”—Medical officer*


Some of the HCWs acknowledged that there were lacunae with follow-up investigations as there was no laboratory facility.


*“We missed only nephropathy (serum creatinine), and we checked diabetes retinopathy through the eye clinic.”—Medical officer*


##### Dependence on Private Laboratories for Investigations (Qualitative Component)

As some of the investigations were not available in the PMCIs, the patients were asked to receive investigations from the private laboratories.


*“We are also doing clinics but once in every 6 months they need to go to private laboratories for renal function investigations”—Medical officer*


When we asked HCWs about the reasons why patients receive investigations from private laboratories, they mentioned that the patients were not aware that all the investigations were available in the PMCIs.


*“People do not know even if we are doing those tests here (PMCI). We need to inform them that we have enough facilities, and we are doing tests. Now this is changing because people are aware that the laboratory is available in our hospitals.”—Medical officer*


##### Need for Visiting Clinic Early in the Morning (Qualitative Component)

The patients mentioned that they had to visit the PMCI early in the morning on the clinic day to avoid long waiting hours.


*“There are patients who arrive by 5–5:30 a.m. So, I get 20–25th place in the queue. The doctor checks my blood pressure and writes medicines quickly. Time is spent waiting to see the doctor and then to obtain medicines. After reaching the pharmacy, they give us medicines quickly”—Patient*


##### Lack of Systems for Referral (Qualitative Component)

The HCWs felt that there were no systems developed at the secondary or tertiary hospitals to hasten the process of consultation of the patients referred from the cluster PMCIs. The HCWs mentioned that because no such systems exist, the referred patients spend a long time at secondary or tertiary hospitals again passing through the common process of going through the OPD gateway. Because the doctors at PMCI would have already assessed the referred patient and would have specified the care needed at hospital, the referred patients could be fast tracked and linked to specific services.


*“A major thing is the waste of time for the patient when they are referred. There is no value for referral there (apex hospital). Here, as only the medical officer and the consultant check them, we send them there (apex hospital). But, the patient has to go through the OPD again there (apex hospital). There is no ‘quota’ given to us to enable a direct referral to a consultant. If a ‘quota’ is given to us, that would be helpful”—Medical officer*


### 3.4. Positive Aspects of NCD Care Provision through PMCI

The people were appreciative about the efforts taken by the PMCIs to screen the individuals through outreach activities in the last 2 years (due to the implementation of PSSP).


*“All (villagers) were happy about this (screening outreach at village) because they did it in the village. We can’t go to the hospital due to our jobs. But as they came to us, we could get everything done.”—Patient*


The patients also appreciated the changes in the NCD care provision through the medical clinics of PMCIs. They felt that the quality of medicines and care provided by the doctor have improved in the last 2 years.


*“Now it is really good compared to how it was before. The medicine is good, the doctor is good, and the staff are also good.”—Patient*


The HCWs also acknowledged that there was increased utilization of the medical clinics with improved care provision.


*“Now the people who were with NCD clinics of XXX hospital (secondary hospital) and private clinics, when they got to know that the dispensary is working well, they all got their clinic books transferred to our clinic… They get all their drugs here. And even the investigations are conducted here… I think the PSSP project has improved the laboratory a lot. They provided the laboratories (apex laboratories to which samples are sent from PMCI for investigations) with machinery and human resources, so now the laboratory has the potential to perform the investigations for all the hospitals allocated to it.”—Medical officer*


## 4. Discussion

This is the first study from Sri Lanka to comprehensively assess the completeness of screening for NCD risk factors, linkages for further care in the HLC or medical clinic, and quality of care provided in the PMCIs supported by PSSP. Since the implementation of PSSP, over 20,000 individuals were screened for NCD risk factors, and this was appreciated by HCWs and the general public. These two groups also felt that utilization of NCD services and quality of care provision had improved under the project. However, during the assessment, certain gaps in performance indicators and challenges were noted in screening, linkage, and care provision.

This operational research had certain strengths. Firstly, the findings from this study conducted within the programmatic setting reflect the real-world situation of NCD care provision in PMCIs. Secondly, data triangulation from various registers and records helped us to deduce indicators which are not routinely monitored under the project, thus providing a roadmap for monitoring and evaluation of NCD care provision in future. Thirdly, as PMCIs were selected from each province of Sri Lanka and we had a reasonable sample size for each step of the assessment, the findings are generalizable. Lastly, qualitative interviews with HCWs, program managers, and patients provided an in-depth insight into the reasons for the gaps identified in the quantitative assessment.

There are a few limitations to be considered when interpreting the results. Firstly, due to deficiency in the recording and reporting system (lack of electronic PHRs) under the program, we were not able to conduct cohort-wise analysis for describing the care cascade of individuals screened for NCD risk factors. Secondly, nonavailability of electronic PHRs and the hard copies of the PHR books in the PMCI meant we had to contact the patients telephonically to assess their clinic books for extracting data on quality of NCD care. We could contact only those with contact details available, who might represent a relatively better educated, affluent, or younger population. Thus, we might have overestimated the proportion that underwent monitoring for complications. Thirdly, unstructured documentation of monitoring for complications in the clinic books might have affected the estimation of the proportion that underwent monitoring for complications. The low percentage calculated could be due to a mixture of both poor documentation and poor compliance with guidelines. Lastly, we were unable to conduct qualitative interviews among those availing care from the private clinics to understand the reasons for their preference. This was because of COVID-19 related travel restrictions and unavailability of details of individuals availing care at private clinics.

More than 20,000 individuals were screened for NCD risk factors, but this accounted for only 23% of all individuals eligible for screening (35 years and above) in the nine PMCIs. The target population, especially males, did not consider screening for NCD risk factors beneficial for themselves, in spite of screening being conducted through outreach activities for ease of access. Similar experiences have been reported in NCD screening efforts from other LMICs [13,32]. The COVID-19 pandemic also hindered the screening activities due to diversion of existing staff and resources to support the COVID-19 response and pandemic-related movement restrictions limiting outreach activities. Such disruptions in service delivery in primary care settings have also been reported globally [33,34,35]. There is a need for scaling up screening by reestablishing outreach activities, as well as leveraging on improved health literacy due to the pandemic for motivating people to utilize screening opportunities. A house-to-house screening strategy through trained community volunteers, which has been shown to be successful in similar settings, could also be considered to accelerate the achievement of the targets [36,37].

Among those screened, only half had total cholesterol measured. This was largely due to the nonavailability of cholesterol strips for investigating total cholesterol with point-of-care (POC) testing devices. Nonavailability of consumables for POC kits leading to underutilization has also been reported in other studies [38,39,40]. A needs assessment exercise should be conducted to estimate demand, and accordingly strengthen the supply chain to ensure an uninterrupted supply of the POC consumables [38]. Only one out of five patients screened had their serum creatinine documented. As the sample collection and transportation systems are not always working effectively and there is no POC test for serum creatinine, it could be dropped from the panel of investigations under screening for NCD risk factors. Instead, this test could be considered only for those diagnosed with hypertension or diabetes, who are considered to be at risk for chronic kidney disease [41]. If the project plans to continue universal screening for serum creatinine, then the availability of the investigation should be improved.

There were deficiencies in the documentation of referral details, with referrals for almost half of the screened individuals not documented. Even among those with documented referral, about one-third were incorrectly referred to HLC or the medical clinic. Moving forward, the screening details should be entered directly into the electronic HMIS module to ensure efficiency and correctness of referral. A decision support system can be integrated with the HMIS to autogenerate referral recommendations based on the entered screening details. This would largely reduce the human error in referrals, as has been evidenced in other settings [42].

In two PMCIs, there were no data sources (registers, health records, or database) for assessing linkage of screened individuals with HLC or medical clinics. In PMCIs where data were available, about 45% of those eligible for referral to a medical clinic were successfully enrolled into care, and 12% of those eligible for follow-up at HLC made at least one follow-up visit. HCWs perceived the lack of systems to track individuals post screening as one of the limitations in linking eligible patients for further care as has been reported in other LMICs [43,44]. To tackle this, there is a need to use electronic PHRs and enhance the utility of HMIS with built-in indicators to monitor the referral and follow-up. Furthermore, the HMIS can be used to generate the due lists for follow-up and list of drop-outs for retrieval actions [7]. As reported from other studies in similar settings, the patients’ preference for private providers was another reason cited for attrition between referral and enrolment at a medical clinic [45,46]. Further qualitative research among those seeking care from private providers could shed light on the reasons for such preferences.

There was a general perception among the HCWs and the patients that the NCD care provision had improved in PMCI with implementation of PSSP. However, only about one-third of the individuals with diabetes underwent blood glucose assessment as per guidelines and about two-thirds of people with hypertension had their blood pressure monitored during their last visit. The lipid profile (<20%), serum creatinine (<10%), ECG (<5%), fundus examination (<2%), and foot examination (<1%) were hardly performed or documented in the previous year among the patients. Such suboptimal performance in the process of care indicators has been reported from other LMICs [47,48,49,50]. Interviews with patients revealed that they were unaware of the need for monitoring for complications using these investigations, as reported elsewhere [51]. Even when the investigations were performed, the majority received them from private facilities. There is a need to strengthen NCD care by improving the laboratory services in the PMCIs. To improve the compliance with monitoring for complications, the importance of this aspect must be emphasized during the NCD training of HCWs [47,52]. As part of patient-centered care, patients need to be educated about the process of NCD care including monitoring for complications. Furthermore, a complication monitoring sheet can be introduced in the PHRs or clinic books [47]. An annual clinical audit with similar indicators used in the study has to be planned to serially assess the care provision through the medical clinics [52,53,54].

Only about two out of five diabetes patients had their blood glucose levels under control, and about one in two hypertensive patients had their blood pressure under control. This mirrors findings from other studies in similar settings [47,49,50]. As reported elsewhere, this could be because patients were not compliant with prescribed medications and lifestyle modifications, as mentioned by HCWs [55,56]. Additionally, not all the NCD drugs were available, requiring patients to purchase these from the private pharmacies. The situation was further aggravated by the COVID-19 pandemic as patients experienced challenges in visiting PMCIs, as reported elsewhere [33]. Annual clinical audits should include the proportion of patients who have achieved control as an outcome indicator for evaluating quality of NCD care provision.

Deficiencies in documentation in the form of incomplete data, lack of uniform recording templates, and dependence on paper formats were noted at all levels of the care provision, i.e., screening, linkage to care, and further follow-up. There is an urgent need for optimizing and operationalizing existing HMIS to improve recording and reporting. Until the HMIS is well established, there is a need to meticulously maintain the paper-based HLC follow-up registers, medical clinic register, and a copy of PHR books in standardized formats to audit the care cascade of individuals screened in the PMCI. Checking for completeness of documentation in the paper-based records during their monitoring visits needs further strengthening. Adopting supportive supervision techniques and educating the HCWs on the performance indicators and their importance in improving NCD care provision are important.

## 5. Conclusions

This first comprehensive assessment of NCD care provision in the PMCIs supported by PSSP showed that the HCWs and general public felt the quality of NCD care provision had improved under the project. However, the project was not able to achieve screening coverage targets (23% against target of 50%). Gaps existed in the documentation of screening results, referral details, and linkages to care. Follow-up investigations such as blood glucose, lipid profile, and serum creatinine were not performed as per guidelines in all diabetes/hypertension patients. This was due to existing programmatic challenges such as the inconsistent use of HMIS, suboptimal laboratory services, and nonavailability of drugs, which were further compounded by the COVID-19 pandemic situation. Priority actions include conducting outreach NCD screening activities, improving the utility of existing HMIS for efficient linkage to care and continued tracking, strengthening laboratory facilities, and drug supply. There is a need for enhancing monitoring and supervision mechanisms to ensure adherence to guidelines for quality care provision.

## Figures and Tables

**Figure 1 healthcare-11-00202-f001:**
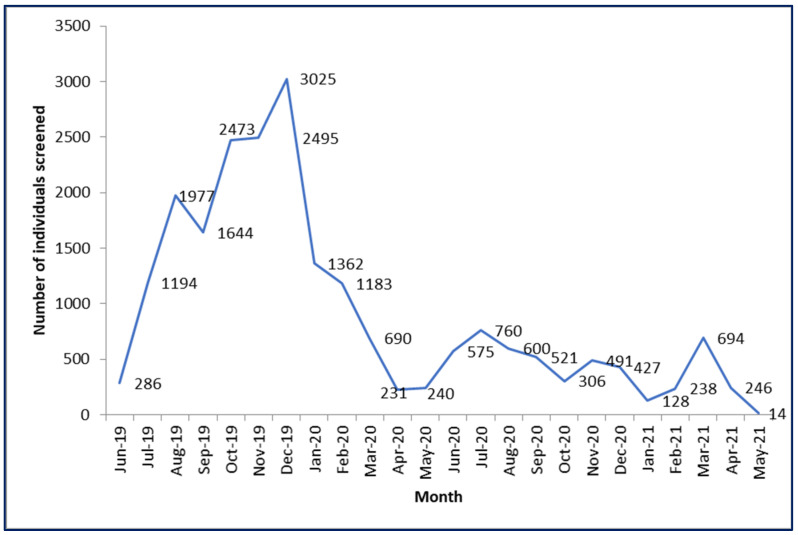
Monthly trend in the aggregate number of individuals screened for NCD risk factors in the nine selected PMCIs supported by the PSSP in Sri Lanka from June 2019 to May 2021. Abbreviations: PMCI—primary medical care institution; NCD—noncommunicable diseases; PSSP—Primary Healthcare System Strengthening Project; GN division—Grama Niladhari division, the most peripheral administrative unit in Sri Lanka.

**Figure 2 healthcare-11-00202-f002:**
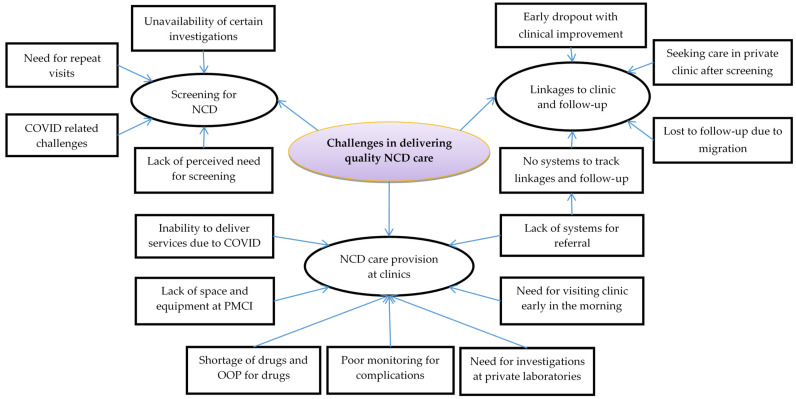
Challenges in delivering quality NCD care in the PMCIs supported by PSSP in Sri Lanka, 2021. Abbreviations: NCD: noncommunicable disease; PMCI: primary medical care institution; PSSP: Primary Healthcare System Strengthening Project; OOP: out-of-pocket expenditure.

**Figure 3 healthcare-11-00202-f003:**
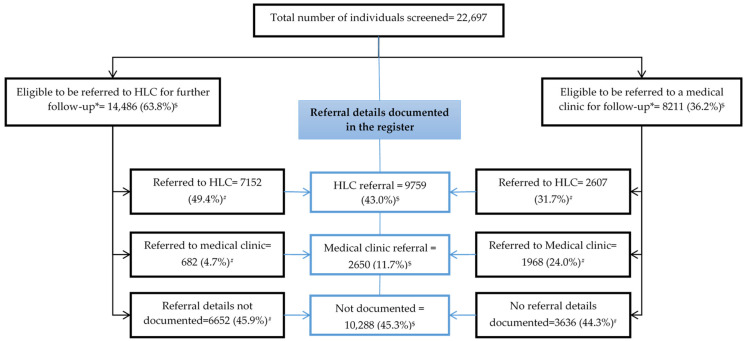
Eligibility for referral and referral status among the individuals screened for NCD risk factors in the selected PMCIs supported by the PSSP in Sri Lanka from June 2019 to May 2021. * Individuals with either “high BP”, “high blood glucose”, “high creatinine”, or 10 year CVD risk ≥ 30% were eligible for referral to medical clinic post screening, and those not eligible to be referred to medical clinic were considered eligible for referral to HLC; ^$^ percentage calculated with total screened as denominator; ^#^ percentage calculated with eligible as denominator. Abbreviations: PMCI—primary medical care institution; NCD—noncommunicable diseases; PSSP—Primary Healthcare System Strengthening Project.

**Figure 4 healthcare-11-00202-f004:**
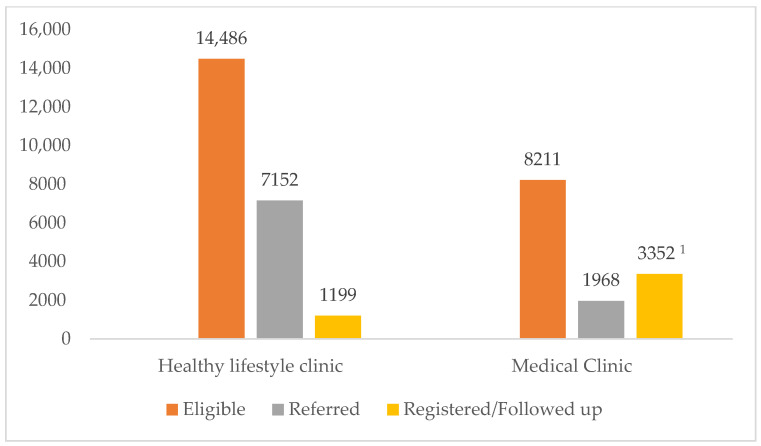
Eligibility for referral, referral status, and follow-up of the individuals who underwent screening for NCD risk factors in the selected PMCIs supported by the PSSP in Sri Lanka from June 2019 to May 2021. Note: Eligible: post NCD risk screening, those who were eligible to be followed up at HLC or referred to medical clinic for further care; Referred: number of individuals who had documentation of suggestion to follow up at healthy lifestyle clinic or referral to medical clinic in the “participant register” after being screened for NCD risk; Registered/Followed up: follow-up at HLC at least once after getting screened for first time and registered for NCD care at medical clinic during study reference period. ^1^ The number registered is the aggregate number of registered individuals extracted from the clinic register, including individuals diagnosed with other NCDs (e.g., asthma, osteoarthritis, and geriatric care), as well as those referred to the clinic from other health facilities (not primarily through screening). Hence, the number registered at the medical clinic is more than the number referred. Abbreviation: PMCI—primary medical care institution; NCD—noncommunicable diseases; PSSP—Primary Healthcare System Strengthening Project.

**Table 1 healthcare-11-00202-t001:** Estimated percentage of individuals aged ≥ 35 years assigned to the PMCI who underwent screening for NCD risk factors ^1^ in the nine selected PMCIs supported by the PSSP in Sri Lanka, 2021.

Primary Medical Care Institution	Individuals Aged ≥ 35 Years in the Identified GN Divisions	Number of Individuals Aged ≥ 35 Years with Documentation Of screening for NCD Risk in the HMIS Database		Number of Individuals Aged ≥ 35 Years Screened for NCD Risk from Assigned GN Division Based on Paper-Based Registers	
		n	(%) ^2^	n	(%) ^2^
PMCI 1	2051	115	(5.6)	436	(21.3)
PMCI 2	6819	1324	(19.4)	1572	(23.1)
PMCI 3	3300	27	(0.8)	1748	(53.0)
PMCI 4	17,145	448	(2.6)	1457	(8.5)
PMCI 5	26,034	952	(3.7)	7624	(29.3)
PMCI 6	1700	402	(23.6)	479	(28.2)
PMCI 7	6994	3033	(44.1)	2953	(42.2)
PMCI 8	7294	443	(6.1)	3075	(42.2)
PMCI 9	15,278	687	(4.5)	871	(5.7)
Total ^3^	86,615	7431	(8.6)	20215	(23.3)

^1^ Includes screening for diabetes, hypertension, behavioral risk factors, and prediction of 10 year CVD risk. ^2^ Percentages were calculated with individuals aged ≥ 35 years in the PMCI as the denominator. ^3^ Cumulative from the nine PMCIs included in the study. Abbreviations: PMCI—primary medical care institution; NCD—noncommunicable diseases; PSSP—Primary Healthcare System Strengthening Project; HMIS—health management information system.

**Table 2 healthcare-11-00202-t002:** Demographic and risk factor profile of the individuals who underwent screening for NCD risk in the selected PMCIs supported by the PSSP in Sri Lanka from June 2019 to May 2021.

Characteristics	PMCI 1	PMCI 2	PMCI 3	PMCI 4	PMCI 5	PMCI 6	PMCI 7	PMCI 8	PMCI 9	Total
	n (%) ^1^	n (%) ^1^	n (%) ^1^	n (%) ^1^	n (%) ^1^	n (%) ^1^	n (%) ^1^	n (%) ^1^	n (%) ^1^	n (%) ^1^
Total	579	1688	1761	2472	8214	542	3043	3501	897	22,697
Age (in years)										
18–24	0 (0)	1 (0.1)	3 (0.2)	41 (1.7)	10 (0.1)	12 (2.2)	1 (0)	71 (2.0)	0 (0)	139 (0.6)
25–34	4 (0.7)	26 (1.5)	10 (0.6)	195 (7.9)	191 (2.3)	41 (7.6)	35 (1.2)	283 (8.1)	3 (0.3)	788 (3.5)
35–44	181 (31.3)	507 (30.0)	488 (27.7)	781 (31.6)	1571 (19.1)	144 (26.6)	783 (25.7)	752 (21.5)	186 (20.7)	5393 (23.8)
45–54	184 (31.8)	431 (25.5)	552 (31.4)	744 (30.1)	1918 (23.4)	133 (24.5)	772 (25.4)	821 (23.5)	256 (28.5)	5811 (25.6)
55–64	161 (27.8)	418 (24.8)	458 (26.0)	468 (18.9)	1813 (22.1)	110 (20.3)	720 (23.7)	838 (23.9)	308 (34.3)	5294 (23.3)
≥65	49 (8.5)	301 (17.8)	248 (14.1)	236 (9.6)	1799 (21.9)	96 (17.7)	730 (24.0)	732 (20.9)	141 (15.7)	4332 (19.1)
Not recorded	0 (0)	4 (0.2)	2 (0.1)	7 (0.3)	912 (11.1)	6 (1.1)	2 (0.1)	4 (0.1)	3 (0.3)	940 (4.1)
Gender										
Male	193 (33.3)	536 (31.8)	0 (0)	778 (31.5)	2601 (31.7)	175 (32.3)	1001 (32.9)	1123 (32.1)	322 (35.9)	6729 (29.7)
Female	386 (66.7)	1152 (68.3)	0 (0)	1693 (68.5)	5593 (68.1)	367 (67.7)	2041 (67.1)	2378 (67.9)	575 (64.1)	14,185 (62.5)
Others	0 (0)	0 (0)	0 (0)	0 (0)	0 (0)	0 (0)	1 (0)	0 (0)	0 (0)	1 (0)
Not recorded	0 (0)	0 (0)	1761 (100)	1 (0)	20 (0.2)	0 (0)	0 (0)	0 (0)	0 (0)	1782 (7.9)
Assigned GN division ^2^										
Yes	396 (68.4)	1506 (89.2)	1716 (97.4)	1456 (58.9)	7718 (94)	404 (74.5)	2988 (98.2)	3041 (86.9)	592 (66.0)	19,817 (87.3)
No	141 (24.4)	94 (5.6)	0 (0)	884 (35.8)	423 (5.2)	14 (2.6)	55 (1.8)	75 (2.1)	23 (2.6)	1709 (7.5)
Not recorded	42 (7.3)	88 (5.2)	45 (2.6)	132 (5.3)	73 (0.9)	124 (22.9)	0 (0)	385 (11)	282 (31.4)	1171 (5.2)
Tobacco Smoking										
Yes	46 (7.9)	169 (10.0)	78 (4.4)	205 (8.3)	554 (6.7)	58 (10.7)	128 (4.2)	278 (7.9)	130 (14.5)	1646 (7.3)
No	507 (87.6)	1299 (77)	1270 (72.1)	2261 (91.5)	6693 (81.5)	482 (88.9)	2912 (95.7)	3195 (91.3)	728 (81.2)	19,347 (85.2)
Not recorded	26 (4.5)	220 (13)	413 (23.5)	6 (0.2)	967 (11.8)	2 (0.4)	3 (0.1)	28 (0.8)	39 (4.4)	1704 (7.5)
Alcohol use										
Yes	71 (12.3)	312 (18.5)	107 (6.1)	443 (17.9)	824 (10.0)	98 (18.1)	310 (10.2)	497 (14.2)	139 (15.5)	2801 (12.3)
No	483 (83.4)	1189 (70.4)	1341 (76.2)	2022 (81.8)	6411 (78.1)	435 (80.3)	2727 (89.6)	2983 (85.2)	727 (81.1)	18,318 (80.7)
Not recorded	25 (4.3)	187 (11.1)	313 (17.8)	7 (0.3)	979 (11.9)	9 (1.7)	6 (0.2)	21 (0.6)	31 (3.5)	1578 (7.0)
Body Mass Index										
<18.5	42 (7.3)	86 (5.1)	125 (7.1)	249 (10.1)	341 (4.2)	70 (12.9)	232 (7.6)	365 (10.4)	87 (9.7)	1597 (7)
18.5–24.9	282 (48.7)	963 (57.1)	869 (49.4)	1150 (46.5)	3522 (42.9)	267 (49.3)	1506 (49.5)	1862 (53.2)	437 (48.7)	10,858 (47.8)
25.0–29.9	183 (31.6)	503 (29.8)	523 (29.7)	792 (32)	3081 (37.5)	152 (28)	980 (32.2)	973 (27.8)	274 (30.6)	7461 (32.9)
≥30.0	71 (12.3)	132 (7.8)	130 (7.4)	279 (11.3)	1270 (15.5)	52 (9.6)	319 (10.5)	299 (8.5)	98 (10.9)	2650 (11.7)
Not recorded	1 (0.2)	4 (0.2)	114 (6.5)	2 (0.1)	0 (0)	1 (0.2)	6 (0.2)	2 (0.1)	1 (0.1)	131 (0.6)
Waist Circumference ^3^										
Normal	197 (34)	932 (55.2)	0 (0)	299 (12.1)	1911 (23.3)	185 (34.1)	900 (29.6)	1788 (51.1)	248 (27.7)	6460 (28.5)
Abnormal	380 (65.6)	737 (43.7)	0 (0)	783 (31.7)	5363 (65.3)	295 (54.4)	2094 (68.8)	1708 (48.8)	195 (21.7)	11,555 (50.9)
Not recorded	2 (0.4)	19 (1.1)	1761 (100)	1390 (56.2)	940 (11.4)	62 (11.4)	49 (1.6)	5 (0.1)	454 (50.6)	4682 (20.6)
Blood Pressure ^4^										
Normal	439 (75.8)	1380 (81.8)	1166 (66.2)	2214 (89.6)	5450 (66.4)	391 (72.1)	2085 (68.5)	2257 (64.5)	590 (65.8)	15,972 (70.4)
High	138 (23.8)	288 (17.1)	100 (5.7)	243 (9.8)	2645 (32.2)	143 (26.4)	956 (31.4)	1200 (34.3)	307 (34.2)	6020 (26.5)
Not recorded	2 (0.4)	20 (1.2)	495 (28.1)	15 (0.6)	119 (1.5)	8 (1.5)	2 (0.1)	44 (1.3)	0 (0)	705 (3.1)
Blood Glucose ^5^										
Normal	487 (84.1)	1405 (83.2)	1512 (85.9)	2048 (82.9)	6894 (83.9)	406 (74.9)	2639 (86.7)	3116 (89)	757 (84.4)	19,264 (84.9)
High	82 (14.2)	265 (15.7)	132 (7.5)	412 (16.7)	1192 (14.5)	58 (10.7)	394 (13)	351 (10)	138 (15.4)	3024 (13.3)
Not recorded	10 (1.7)	18 (1.1)	117 (6.6)	12 (0.5)	128 (1.6)	78 (14.4)	10 (0.3)	34 (1)	2 (0.2)	409 (1.8)
Total Cholesterol ^6^										
Normal	395 (68.2)	609 (36.1)	982 (55.8)	153 (6.2)	948 (11.5)	109 (20.1)	1059 (34.8)	2113 (60.4)	101 (11.3)	6469 (28.5)
High	140 (24.2)	872 (51.7)	676 (38.4)	230 (9.3)	444 (5.4)	213 (39.3)	743 (24.4)	1366 (39)	213 (23.8)	4897 (21.6)
Not recorded	44 (7.6)	207 (12.3)	103 (5.9)	2089 (84.5)	6822 (83.1)	220 (40.6)	1241 (40.8)	22 (0.6)	583 (65)	11,331 (49.9)
Serum Creatinine ^7^										
Normal	172 (29.7)	432 (25.6)	1534 (87.1)	315 (12.7)	38 (0.5)	0 (0)	1 (0)	922 (26.3)	2 (0.2)	3416 (15.1)
High	9 (1.6)	158 (9.4)	102 (5.8)	16 (0.7)	0 (0)	0 (0)	0 (0)	16 (0.5)	0 (0)	301 (1.3)
Not recorded	398 (68.7)	1098 (65.1)	125 (7.1)	2141 (86.6)	8176 (99.5)	542 (100)	3042 (100)	2563 (73.2)	895 (99.8)	18,980 (83.6)
CVD Risk ^8^										
<10%	567 (97.9)	1376 (81.5)	0 (0)	2312 (93.5)	6422 (78.2)	503 (92.8)	2710 (89.1)	2577 (73.6)	836 (93.2)	17,303 (76.2)
10–20%	1 (0.2)	65 (3.9)	0 (0)	32 (1.3)	371 (4.5)	22 (4.1)	121 (4)	184 (5.3)	45 (5)	841 (3.7)
20–30%	1 (0.2)	16 (1)	0 (0)	12 (0.5)	137 (1.7)	7 (1.3)	58 (1.9)	65 (1.9)	6 (0.7)	302 (1.3)
>30%	1 (0.2)	4 (0.2)	0 (0)	6 (0.2)	187 (2.3)	5 (0.9)	52 (1.7)	58 (1.7)	10 (1.1)	323 (1.4)
Not recorded	9 (1.6)	227 (13.5)	1761 (100)	110 (4.5)	1097 (13.4)	5 (0.9)	102 (3.4)	617 (17.6)	0 (0)	3928 (17.3)

^1^ Column percentage calculated with total screened in the PMCI as the denominator. ^2^ Individual screened is from the GN division assigned to the PMCI. ^3^ A waist circumference < 90 for male and <80 for female was considered normal; their counterparts were considered high. ^4^ A systolic blood pressure < 140 mmHg and diastolic blood pressure < 90 mmHg were considered normal; their counterparts were considered high. ^5^ A fasting blood glucose < 126 mg/dL and random blood glucose < 200 mg/dL were considered normal. ^6^ A total cholesterol < 200 mg/dL was considered normal. ^7^ A serum creatinine ≤ 1.2 was considered normal. ^8^ As ascertained and documented in the “participant register”.

**Table 3 healthcare-11-00202-t003:** The percentage of individuals with diabetes and/or hypertension screened for complications as per the guidelines and having the values of screened parameters under control in the nine selected PMCIs supported by the PSSP of Sri Lanka, June 2021.

Investigation/Examination	Eligible ^1^	Tested		Control/Normal ^2^	
	N	n	(%) ^3^	N	(%) ^4^
**Diabetes**					
BP measurement in the last visit	349	199	(57.0)	126	(63.3)
Blood glucose test conducted in the last 3 months	328	117	(35.7)	46	(39.3)
Lipid profile test in the last year	229	41	(17.9)	7	(17.1)
Renal function test in the last year	229	16	(7.0)	12	(75.0)
ECG in the last year	229	10	(4.4)	10	(100)
Foot examination in the last year	229	4	(1.8)	3	(75.0)
Fundus examination	229	2	(0.9)	2	(100)
**Hypertension**					
BP measurement in the last visit	326	228	(69.9)	107	(46.9)
Blood glucose assessment in the last year	225	62	(27.6)	49	(79.0)
Lipid profile test in the last year	225	28	(12.4)	7	(25.0)
Renal function test in the last year	225	7	(3.1)	7	(100.0)
ECG in the last year	225	13	(5.8)	12	(92.3)
Fundus examination	225	1	(0.4)	1	(100)

^1^ All the individuals with diabetes were eligible for BP measurement during their last visit; the individuals with diabetes registered for care 3 months prior to date of data extraction were eligible for undergoing blood glucose tests at least once in the last 3 months; the individuals with diabetes registered for care 12 months prior to date of data extraction were eligible for undergoing all the other investigation at least once in the last year. All the individuals with only hypertension were eligible for BP measurements during their last visit; the individuals with only hypertension registered for care 12 months prior to date of data extraction were eligible for undergoing all the other investigation at least once in the last year. ^2^ Systolic blood pressure < 140 mmHg and diastolic blood pressure < 90 mmHg were considered to be under control; fasting blood glucose < 126 mg/dL or random blood glucose < 200 mg/dL was considered to be under control; total cholesterol < 200 mg/dL, LDL < 100 mg/dL, HDL ≥ 40 mg/dL, and triglycerides < 150 were considered a lipid profile under normal limits; serum creatinine < 1.3 mg/dL was considered a renal function test under normal limits. For ECG, foot examination, and fundus examination, documentation of the normal status by the examiner was considered to represent normal control. ^3^ Percentage calculated with number eligible for testing as the denominator. ^4^ Percentage calculated with number tested as the denominator. Abbreviations: PMCI—primary medical care institution; PSSP—Primary Healthcare System Strengthening Project; ECG—electrocardiography.

**Table 4 healthcare-11-00202-t004:** Place of investigation/examination among diabetes and/or hypertension patients undergoing screening for complications as per guidelines in the nine selected PMCIs supported by PSSP of Sri Lanka, 2021.

Investigation/Examination	Total	Place of Investigation			
	N	PMCI	Public ^1^	Private	Not Recorded
		n (%) ^2^	n (%) ^2^	n (%) ^2^	n (%) ^2^
Blood Glucose	179	78 (43.5)	26 (14.5)	51 (34.2)	14 (7.8)
Lipid Profile	69	26 (37.7)	11 (15.9)	26 (37.8)	8 (8.7)
Renal Function Test	23	5 (21.7)	8 (34.8)	7 (30.4)	3 (13.1)
ECG	23	18 (78.3)	3 (13.0)	2 (8.7)	0 (0.0)
Foot examination	4	4 (100)	0 (0)	0 (0)	0 (0)
Fundus	3	3 (100)	0 (0)	0 (0)	0 (0)

^1^ Any public health facility other the PMCI where the patient is registered for care. ^2^ The percentages are calculated with row total as the denominator. Abbreviations: PMCI—primary medical care institution; PSSP—Primary Healthcare System Strengthening Project; ECG—electrocardiography.

## Data Availability

Requests to access these data should be sent to the corresponding author.

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
