# Peer review of "Outcomes and Challenges in Noncommunicable Disease Care Provision in Health Facilities Supported by Primary Health Care System Strengthening Project in Sri Lanka: A Mixed-Methods Study"

_healthcare, 2023, doi:10.3390/healthcare11020202_

Round 1
Reviewer 1 Report
I believe the research has a great potential in conveying the messages about the risks of NCD to the public and provide some room for better advocacy to the current program, especially for LMICs and its related challenges. It would be better if the authors could pinpoint concrete facts of the outcomes and challenges in the abstract (as the title of the paper); without putting too much detailing on the statistics outcomes. What does the statistics results say? What are the end outcomes that show the challenges? I have not seen the link clearly in the text; therefore, in the discussion and conclusion, this link should be mentioned clearly. I would also strongly recommend the link between the quantitative and qualitative methods and their results be better illustrated. The authors have mentioned, explanatory mixed method, true; but what is their connection? I would also strongly recommend the criteria of the respondents for qualitative research/method be explained (see line 172-177). How do you select these respondents for the interview? What is the basis? Is there a link from the quantitative results? At the conclusion, usually the authors provide a glimpse on how to overcome the challenges that are born from the research results; for better healthcare program of the reduction of NCD risks.
Author Response
We thank the reviewer very much for the time and effort spent in reviewing our paper. This is much appreciated.
I believe the research has a great potential in conveying the messages about the risks of NCD to the public and provide some room for better advocacy to the current program, especially for LMICs and its related challenges. It would be better if the authors could pinpoint concrete facts of the outcomes and challenges in the abstract (as the title of the paper); without putting too much detailing on the statistics outcomes. What does the statistics results say? What are the end outcomes that show the challenges?
Response: Thank you for this observation. We have edited the abstract and have reduced the detailing on statistical content. We have now tried to present the challenges of NCD care provision (qualitative component) along with the key end outcomes at each step (quantitative findings). However, given the restriction of 200 words for the abstract, we have not been able to detail all the challenges from the main narrative in the abstract. Changes made in lines 28 to 36 (simple mark-up view in document with track changes).
I have not seen the link clearly in the text; therefore, in the discussion and conclusion, this link should be mentioned clearly. I would also strongly recommend the link between the quantitative and qualitative methods and their results be better illustrated. The authors have mentioned, explanatory mixed method, true; but what is their connection?
Response: Thank you for this comment. We have made the following changes in an attempt to show the link between quantitative and qualitative methods
- In the methods sections, we have now mentioned how the quantitative findings guided our qualitative inquiry (Lines 248-251, simple mark-up view in document with track changes)
- Results: We had initially presented the quantitative and qualitative findings separately. We have now reorganised this entire section. For each step, we present the gaps identified quantitatively; followed by an exploration of possible reasons for these gaps which were identified using qualitative methods.
- In the Discussion section and Conclusions, we have drawn inferences about each of the identified gaps based on our qualitative exploration. The recommendations to address the gaps have been made in line with the qualitative findings.
I would also strongly recommend the criteria of the respondents for qualitative research/method be explained (see line 172-177). How do you select these respondents for the interview? What is the basis? Is there a link from the quantitative results?
Response: Thank you for this observation. We have explained the selection of respondents for qualitative interviews in Lines 178-185 (simple mark-up view in document with track changes).
At the conclusion, usually the authors provide a glimpse on how to overcome the challenges that are born from the research results; for better healthcare program of the reduction of NCD risks.
Response: Thank you for this suggestion. We have included key recommendations in the Conclusion section (Lines 847-856, simple mark-up view in document with track changes).
Reviewer 2 Report
1. This manuscript conducted an explanatory mixed-methods study to assess the completeness of screening for NCD risk, linkage to care, and outcomes of diabetes/hypertension care at nine selected PMCIs. However, most of the analysis methods focus on the simple percentage comparison without other complicated analysis methods for the multi-indicators. Although the authors give some indicators from different aspects, no multi-index evaluation method is presented to give a comprehensive analysis result.
2. The challenges referring to 15 categories are presented as text descriptions, but the authors refer to 63 codes related to challenges, which cannot be reflected in using these codes to conduct the analysis further. Too much description of the related challenges without giving the related resolutions and this manuscript lacks an academic presentation.
Author Response
We thank the reviewer very much for the time and effort spent in reviewing our paper. This is much appreciated.
This manuscript conducted an explanatory mixed-methods study to assess the completeness of screening for NCD risk, linkage to care, and outcomes of diabetes/hypertension care at nine selected PMCIs. However, most of the analysis methods focus on the simple percentage comparison without other complicated analysis methods for the multi-indicators. Although the authors give some indicators from different aspects, no multi-index evaluation method is presented to give a comprehensive analysis result.
Response: Thank you for this comment. We appreciate the comment from the reviewer, but would like to differ from the reviewer’s viewpoint here. We strongly believe that each step in the cascade of care outlined in this manuscript (screening, linkage to care and follow-up) is a distinct process. The challenges in each of these steps are different in this context and therefore need to be addressed separately. When solutions are identified and implemented, subsequent evaluations will also need to look at the indicators in each of these steps separately. The objective of this study was to identify gaps and associated reasons which would enable policy makers to make informed decisions for improving NCD care provision. Using a complicated analysis method for the multi-indicators and developing a composite index will not be well understood or deemed to be relevant for the programme at this stage. We would like to restrict ourselves to the objectives set out at the end of the Introduction and which were developed in consultation with the programme managers in Sri Lanka.
The challenges referring to 15 categories are presented as text descriptions, but the authors refer to 63 codes related to challenges, which cannot be reflected in using these codes to conduct the analysis further. Too much description of the related challenges without giving the related resolutions and this manuscript lacks an academic presentation.
Response: We would like to clarify our terms “categories” and “codes”. As part of qualitative analysis, we first coded the information by reviewing the transcripts of the interviews. Coding is the process of labelling and organizing the qualitative data. Once the codes were generated, codes which concerned a specific challenge were grouped together as a “category”. For example, codes like ‘inability of individuals to visit hospital for screening during the COVID pandemic due to lack of transportation’, ‘inability to conduct outreach activities due to COVID restrictions’, ‘diversion of staff to COVID response activities’ and ‘designation of PMCIs as COVID treatment facilities’ were all combined to form a category called ‘COVID related challenges’ under the theme of ‘Challenges for screening’. The investigators have all been trained in qualitative research, they are experienced in this research discipline and have published extensively on qualitative research in the past. We have also adhered to international guidelines from credible literature on how to conduct and report on qualitative research (COREQ guidelines, Qualitative inquiry and research design by Poth and Creswell).
As previously stated, the mandate of this research was to identify gaps at different steps of NCD care provision and the associated reasons for these gaps within a programmatic setting. Since, exploration of solutions was not part of the study, we have not ventured to do so in the Results section. However, some of the possible solutions, which could be considered in view of the challenges identified from the perspectives of healthcare providers and patients, have been proposed in the Discussion and Conclusion sections.
Reviewer 3 Report
Please move the following sentences to the discussion section.
31-32
30 Healthcare providers attributed poor coverage to lack of public awareness and COVID-19 related 31 service disruptions.
34-36
Lack of robust tracking mechanisms and preference for private providers contributed to post-screening attrition.
39-40
Primary care strengthening needs to address these challenges to ensure successful integration of NCD care within PMCIs. 40
479-486
3.2.1.1. Lack of perceived need for screening 479
The health care workers (HCWs) felt that the general public did not perceive the 480 importance of the screening even when it was explained to them, especially those who felt 481 apparently healthy. 482 “I feel that they may not have perceived the true value of the screening process. And I'm not 483 quite sure when they're invited through the phone, whether the weight that it should carry about 484 the importance of the screening was clearly conveyed to them... And then for people who are not 485 with any illnesses. I mean they don't get something big in return.” – Medical officer 486
487-491
3.2.1.2. Non-availability of certain investigations 487
The HCWs complained that some recommended investigations like serum creatinine 488 and total cholesterol were not available during the screening activities. Sample collection 489 and transportation mechanisms for receiving tests in the apex laboratories were not adequately established. 491
514-516
Then, there is the probability of missing a few patients. That means, 514 even though the patient joined in the screening activity, he (patient) might not like to spend another 515 day to come and collect results due to his personal reason.”-Medical officer 516
517-520
3.2.1.4. COVID related challenges 517
The HCWs felt that screening activities reduced drastically with the onset of the 518 COVID-19 pandemic. This was due to a shift in focus of HCWs towards the COVID-19 519 response and also due to the inability to conduct outreach activities for screening.
532-541
3.2.2.1. No systems to track linkages and follow-up 532
The HCWs felt that the systems for tracking with either a paper-based register or 533 electronic HMIS were not well established for tracking the individuals once they were 534 screened for NCD risk factors. The PHRs were available only with the patients and electronic PHRs were not available. Thus, there was no opportunity to derive a list of individuals not linked at medical clinic after screening. Similarly, there were no registers to document the registration and follow-up details of patients seeking care in the medical clinic. 538 “Actually, we are not maintaining registers for documenting follow-up details of patients 539 followed up in the medical clinic. We will do it in future. Once medical clinic details are entered in 540 the HMIS, we will be able to track the patients.
The following summary text should be moved to the discussion.
814-821
This first comprehensive assessment of NCD care provision in the PMCIs supported 814 by PSSP showed that the HCWs and general public felt the quality of NCD care provision 815 had improved under the project. However, the project was not able to achieve screening 816 coverage targets and ensure optimal NCD care as envisioned. This was due to existing 817 programmatic challenges like the inconsistent use of HMIS, sub-optimal laboratory ser- 818 vices and non-availability of drugs, which were further compounded by the COVID-19 819 pandemic situation. Moving forward, there is an urgent need to address these challenges 820 to improve NCD care provision. 821
New " 5.Conclusion "
Please replace the following sentences as a new "Conclusion"
88-95
Two years into the implementation of PSSP (The Primary Health Care System Strengthening Project), we conducted the first systematic assessment of the status of implementation of the project in the country to generate evidence for 89 guiding its future scale-up. We used quantitative and qualitative techniques of inquiry to 90 assess facility level infrastructure for care provision, coverage of empanelment, process of 91 NCD care provision and patient experience at PMCIs supported by PSSP. In this paper, 92 we report the findings of assessment of the coverage and completeness of screening for 93 NCD risk factors, linkage with care and outcomes of care among those enrolled for diabetes and/or hypertension care at nine PMCIs in Sri Lanka.
30
Against a screening coverage target of 50% among individuals aged ≥35 years, PMCIs achieved 23%. 30
32-34
Total cholesterol (50%) and serum creatinine (84%) were not recorded/done in 32 those screened. Referral information based on screening results was not available for 45% of those 33 screened. Post-screening, 45% of those eligible for NCD care at medical clinics were registered.
。
36-39
Follow-up investigations like blood glucose (36%), lipid profile (<20%) and serum creatinine (<10%) were not done as per guidelines in all diabetes/hypertension patients due to non-availability of investigations at PMCI. Glycemic and blood pressure control was noted in 39% of diabetes and 47% of hypertension patients, respesctively.
Author Response
We thank the reviewer very much for the time and effort spent in reviewing our paper. This is much appreciated.
Please move the following sentences to the discussion section.
31-32: 30 Healthcare providers attributed poor coverage to lack of public awareness and COVID-19 related 31 service disruptions.
34-36: Lack of robust tracking mechanisms and preference for private providers contributed to post-screening attrition.
39-40: Primary care strengthening needs to address these challenges to ensure successful integration of NCD care within PMCIs. 40
479-486:
3.2.1.1. Lack of perceived need for screening 479
The health care workers (HCWs) felt that the general public did not perceive the 480 importance of the screening even when it was explained to them, especially those who felt 481 apparently healthy. 482 “I feel that they may not have perceived the true value of the screening process. And I'm not 483 quite sure when they're invited through the phone, whether the weight that it should carry about 484 the importance of the screening was clearly conveyed to them... And then for people who are not 485 with any illnesses. I mean they don't get something big in return.” – Medical officer 486
487-491
3.2.1.2. Non-availability of certain investigations 487
The HCWs complained that some recommended investigations like serum creatinine 488 and total cholesterol were not available during the screening activities. Sample collection 489 and transportation mechanisms for receiving tests in the apex laboratories were not adequately established. 491
514-516
Then, there is the probability of missing a few patients. That means, 514 even though the patient joined in the screening activity, he (patient) might not like to spend another 515 day to come and collect results due to his personal reason.”-Medical officer 516
517-520
3.2.1.4. COVID related challenges 517
The HCWs felt that screening activities reduced drastically with the onset of the 518 COVID-19 pandemic. This was due to a shift in focus of HCWs towards the COVID-19 519 response and also due to the inability to conduct outreach activities for screening.
532-541
3.2.2.1. No systems to track linkages and follow-up 532
The HCWs felt that the systems for tracking with either a paper-based register or 533 electronic HMIS were not well established for tracking the individuals once they were 534 screened for NCD risk factors. The PHRs were available only with the patients and electronic PHRs were not available. Thus, there was no opportunity to derive a list of individuals not linked at medical clinic after screening. Similarly, there were no registers to document the registration and follow-up details of patients seeking care in the medical clinic. 538 “Actually, we are not maintaining registers for documenting follow-up details of patients 539 followed up in the medical clinic. We will do it in future. Once medical clinic details are entered in 540 the HMIS, we will be able to track the patients.
Response: Thank you for these suggestions. However, these statements and sentences are all part of our qualitative results and as such we feel strongly that they should remain in the Results section. We have adhered to the internationally recognised “Consolidated criteria for reporting qualitative research (COREQ) guidelines” for reporting, according to which quotations and findings (codes, categories and themes) are reported under the Results section. Therefore, we would prefer to retain these sentences in the Results Section as has also been emphasised by Reviewer 1. We hope the reviewer can accept our stance here.
The following summary text should be moved to the discussion.
814-821: This first comprehensive assessment of NCD care provision in the PMCIs supported 814 by PSSP showed that the HCWs and general public felt the quality of NCD care provision 815 had improved under the project. However, the project was not able to achieve screening 816 coverage targets and ensure optimal NCD care as envisioned. This was due to existing 817 programmatic challenges like the inconsistent use of HMIS, sub-optimal laboratory ser- 818 vices and non-availability of drugs, which were further compounded by the COVID-19 819 pandemic situation. Moving forward, there is an urgent need to address these challenges 820 to improve NCD care provision. 821
Response: Thank you for this suggestion. A similar message, however, has already been conveyed in the first paragraph of Discussion. Each of the components are detailed in the later paragraphs of Discussion. Thus, to keep Conclusions as a stand-alone section (journal author guidelines) and to avoid repetition of the messages in Discussion section, we would prefer to retain the content as it is. Again, we hope the reviewer can accept our stance here.
New " 5.Conclusion ": Please replace the following sentences as a new "Conclusion"
88-95: Two years into the implementation of PSSP (The Primary Health Care System Strengthening Project), we conducted the first systematic assessment of the status of implementation of the project in the country to generate evidence for 89 guiding its future scale-up. We used quantitative and qualitative techniques of inquiry to 90 assess facility level infrastructure for care provision, coverage of empanelment, process of 91 NCD care provision and patient experience at PMCIs supported by PSSP. In this paper, 92 we report the findings of assessment of the coverage and completeness of screening for 93 NCD risk factors, linkage with care and outcomes of care among those enrolled for diabetes and/or hypertension care at nine PMCIs in Sri Lanka.
Response: Thank you for the suggestion. We believe that these sentences provide the rationale for the study and set the context of the assessment for the reader. We kindly request that these sentences be retained in the Introduction in line with the Strengthening the Reporting of Observational Studies in Epidemiology (STROBE) and COREQ guidelines.
30: Against a screening coverage target of 50% among individuals aged ≥35 years, PMCIs achieved 23%. 30
32-34: Total cholesterol (50%) and serum creatinine (84%) were not recorded/done in 32 those screened. Referral information based on screening results was not available for 45% of those 33 screened. Post-screening, 45% of those eligible for NCD care at medical clinics were registered.
36-39: Follow-up investigations like blood glucose (36%), lipid profile (<20%) and serum creatinine (<10%) were not done as per guidelines in all diabetes/hypertension patients due to non-availability of investigations at PMCI. Glycemic and blood pressure control was noted in 39% of diabetes and 47% of hypertension patients, respectively.
Response: Thank you for the suggestions. We have edited the Conclusion section to incorporate these suggestions, as well as some suggestions that were received from Reviewer 1 (Lines 846-850, simple mark-up view in document with track changes).
Reviewer 4 Report
Overall well written paper and organized study reflecting the importance of early screening and to evaluate infrastructure needed in managing non- communicable diseases given the growing burden in developing countries and throughout the world. My only suggestion is try to include more population to see if the authors as described can hit the 50% screening target.
Author Response
Overall well written paper and organized study reflecting the importance of early screening and to evaluate infrastructure needed in managing non- communicable diseases given the growing burden in developing countries and throughout the world.
Response: Thank you for appreciating the manuscript. We authors also feel that learnings from this will be useful in planning similar NCD related initiative in low-and-middle-income countries.
My only suggestion is try to include more population to see if the authors as described can hit the 50% screening target.
Response: Thank you for the suggestion. We have included 86615 individuals for assessing the screening coverage and about 20215 (23.3%, 95% CI- 23.0-23.6) got screened as opposed to target of 50%. As you can can appreciate, we have huge sample size, which reflected in the narrow confidence interval (95% CI). We have updated this confidence interval in abstract (line number-31) and also in results section (Line number-303). There is no scope for increasing the sample size at this stage as the operational research study for evaluating the extent of programmatic implementation of screening was completed in the end of 2021. We hope that inclusion of confidence interval is sufficient to highlight the huge sample size in the study and the coverage would remain same even with inclusion of larger sample.
Round 2
Reviewer 2 Report
My comments haven't been addressed.
Author Response
My comments haven't been addressed.
Response: Thank you for the comment. As described, we have tried our best to address all the comments in our response document.